# OpenReview forum: "BoNBoN Alignment for Large Language Models and the Sweetness of Best-of-n Sampling"
_NeurIPS.cc/2024/Conference — NeurIPS 2024 poster_

### Official Review · Reviewer_ySG3 · 2024-06-28

**Soundness:** 3
**Presentation:** 3
**Contribution:** 2
**Rating:** 5
**Confidence:** 2

**Summary:**

This paper studies aligning samples from LLMs with human preferences using best-of-$n$ (BoN) sampling methods.
While BoN methods can yield more desirable outputs without changing off-target behavior, they are computationally expensive since they require $n$ samples from the LLM for every prompt.
To address this, the paper proposes BoNBoN alignment, which aims to mimic the sample distribution of BoN without generating $n$ samples.
The authors demonstrate that BoNBoN alignment achieves a high win-rate while minimally changing off-target behavior.

**Strengths:**

### Clarity and Quality
This paper is generally well-written and easy to understand.
The idea itself sounds intuitive and the technical results I checked seem to be correct although some proofs should be modified.

### Significance
It is worth noting that I am not specialized both in LLM and generative model.
Therefore, while I think the overall claim sounds reasonable, I will not judge how this work is novel and has impact in this field.

**Weaknesses:**

See Questions.

**Questions:**

(Q1) For optimal policy in (3.4), since $Q\_x$ is a CDF of $r(x, Y_0)$, it is a monotonically increasing function, which implies that
$$\begin{equation}
\arg\max\_{\pi} \mathbb{E}[Q\_x(r(x, y))] = \arg\max\_{\pi} \mathbb{E}[r(x, y)].
\end{equation}$$
Therefore, the optimal policy can be written in a very similar formulation to (2.8).
Why do we need to consider $Q\_x$ as in (3.5)?

---

While the authors choose $\alpha$ so that the losses from SFT-BON and IPO-BON contribute approximately equally to the total loss, and claim that this is much easier than choosing $\beta$, this remains unclear to me.
When training a fine-tuning model, one will observe empirical losses from SFT-BON and IPO-BON for given prompts.
The empirical BoNBoN alignment objective can then be computed as an internally dividing point between the objectives of SFT-BON and IPO-BON.

(Q2) How to set $\alpha$ before observing the empirical objectives of SFT-BON and IPO-BON? Since $\alpha$ is chosen to balance the terms between them, it seems necessary to know the ratio between them prior to training.

---

Although BON does not require choosing $\beta$, the choice of $n$ implicitly plays a similar role as $\beta$, as shown in Theorems 1 and 2.
Therefore, a naive question would be:

(Q3) how should one choose $n$?

When $n$ is too large, the possible answers of BON for similar questions would converge to one specific answer, resembling a point mass.
In this scenario, the KL-divergence between BON and the optimal distribution could be large, even though the win-rate increases.
This implies that a large $n$ is somewhat equivalent to small $\beta$.

---

### Major comment

The proof of Theorem 1 from lines 448 to 449 needs modification.
The current arguments using **max** and **min** are mathematically incorrect, as the authors have skipped all coefficients, including negative ones.
The arguments should be written in terms of **argmax** and **argmin** or including all coefficient explicitly.

**Limitations:**

### Minor comments
Although this paper is well-written in general, there are some places missing information.

1. What is $y\_w$ and $y\_l$ in (2.6).

2. What is $\sigma$ in (2.7).

3. What is $r^*$ in line 435.

4. What is $D\_{BW}$ in line 472.

---
After rebuttal:
I have increased my score from 4 to 5 as the authors have addressed my concerns.
The reason I am giving a 5 is that I am unable to fully assess the impact of this work within the LLM field.

---

> ### Author Rebuttal · Authors · 2024-08-06
>
> Thank you for your review!
>
> - (A1): Since the expectation also accounts for $x$, the left side and the right side are not equivalent. For different prompts, the distribution of rewards varies. This is fundamental: some prompts are easy for the base model (e.g., “what is 2+2?”) and the optimal alignment is to do nothing. Some prompts have responses with highly variable rewards, and the optimal alignment is to strongly favor the good responses. Having a prompt-specific transformation is what lets us take advantage of this structure.
>
> - (A2): Morally, the point here is that choosing $\alpha$ is more akin to choosing the learning rate than to choosing a parameter governing KL vs reward tradeoff. In theory, it should affect the optimization path, but (ignoring finite sample issues) not the final solution. In principle, to find the optimal KL vs reward tradeoff, one would need to fully fit the model at many distinct values. By contrast, all values of $\alpha$ such that the model converges well should yield the same behavior. So, we only need to find one. In practice, we just use a heuristic to set it based on the losses in the first steps of training.
>
>     We also note that this is not the main point of the paper. It does seem like a useful, material advantage. But the core point of the paper—the optimality and achievability of best-of-$n$ alignment—stands irrespective.
> - (A3): In principle, one could choose $n$ large enough to observe reward hacking type behavior. However, the win rate increases relatively slowly for $n>10$ (at which point the win rate is already 90%) and the KL divergence increases very slowly. This win rate already suffices to beat existing contrastive approaches with very small KL drift. So, in practice, choosing $n$ around 10 seems to be a sweet spot.
>
>     It is an interesting question how to adapt the procedure in situations where more extreme model modification is desirable. We suspect that an iterated best-of-$n$ procedure using multiple rounds of BonBon would be effective. However, this is a direction for future work.
> - For major comment: Thanks for pointing this out! We will change the proof in the updated version.
> - For minor comment:
>   - 1-2: $y_w$ and $y_l$ are the win and lose response of the prompt $x$. $\sigma$ is the sigmoid function. Their definitions will be added after the equations in the updated version.
>   - $r^*$ is $r$ and $D_{BW}$ stands for the (prompt, best response, worst response) dataset. These notations are typos left over from an earlier version. Thanks for pointing them out. We will fix these typos in the revised version.

---

> > ### Author Response · Authors · 2024-08-12
> >
> > Thank you again for your review! Do you have any further questions? If we have resolved your primary concerns, we would greatly appreciate it if you could consider raising the overall rating. If there are any additional improvements we can make to earn a higher score, we would be more than happy to address them.

---

> > > ### Comment · Reviewer_ySG3 · 2024-08-12
> > >
> > > Sorry for the late reply.
> > > Thanks for the authors' response.
> > > I will raise my score to 5.

---

### Official Review · Reviewer_mr5a · 2024-07-13

**Soundness:** 4
**Presentation:** 3
**Contribution:** 3
**Rating:** 7
**Confidence:** 3

**Summary:**

The paper describes theoretical results about the Best-of-n sampling procedure in LLM inference. To reduce the computational cost of the procedure, authors develop a novel finetuning method called BoNBoN. Experiments on dialog generation and text summarization show that BoNBoN achieves a higher win-rate for the same KL divergence from the reference model compared to SFT, DPO, and IPO applied on BoN samples.

**Strengths:**

* The paper gives a concise introduction to RLHF and DPO and views the reward-optimal conditional probability as an exponential tilting of the reference model's conditional probability. This motivates the $f_x$-aligned optimal policy.
* The paper analytically solves for the optimal policy at a given KL divergence value from the reference model, and shows that the Best-of-n policy approximates it well.
* To remove the need for sampling n times, the idea of BoNBoN is proposed. Experimental results show that it attains a better win-rate vs KL divergence tradeoff compared to other approaches.

**Weaknesses:**

* It would interesting to plot the theoretically optimal tradeoff in Theorem 1 on the plot in Figure 3, to better visualize the performance of different methods across a range of different KL values.
* More description about Fig 3 would be helpful. What was the value of n used to obtain the BoN sampling operating point? It would be more complete to see different operating points of BoNBoN for different values of n.

**Questions:**

* IPO-BoN is presented as making use of more than just the winning sample (specifically, it uses the best and worst samples in a contrastive objective). It naturally raises the question of using other pairs in the loss as well. Do the authors think that taking more than one pair in the loss could be analyzed in a similar manner, or is it not expected to yield an advantage empirically?

---

> ### Author Rebuttal · Authors · 2024-08-06
>
> Thank you for your review!
>
> First, thanks for all your suggestions about Figure 3! To make it more informative, we will add the theoretical optimal line in the figure and more detailed information in the caption. For the question in the second point, the $n$ in Fig 3 is 8 (we mentioned in Section 5.1 Experimental Setup) and we will add this to the caption. We chose 8 since the win rate of 8 is large enough (nearly 90%) to be favorably competitive with existing contrastive methods. This allows us to compare the win-rate vs KL frontier in the relevant regime. We will also add comparisons for $n=2,\cdots,8$ to the camera ready plots.
>
> For your question: we’ve tried best vs other samples (not the worst one) as pairs for IPO and found that the best vs worst pairs have the most satisfying performance. It would be an interesting direction for future work to find a way to augment the training objective to consume multiple distinct pairs. It is quite possible this could further improve performance. However, we think the simplicity of the BonBon approach is desirable for this paper, since it makes it clear that the advantage is just about best-of-$n$ rather than, e.g., increasing the effective amount of data used for contrastive alignment.

---

### Official Review · Reviewer_PJN3 · 2024-07-13

**Soundness:** 2
**Presentation:** 1
**Contribution:** 3
**Rating:** 5
**Confidence:** 2

**Summary:**

They claim that best-of-n is an optimal policy with respect to the tradeoff between win rate and the KL divergence. Based on the analysis they propose a strategy to train a model so that it gets to policy similar to the BoN policy.

**Strengths:**

The research question is interesting. BoN and the other learning-based alignment algorithms are currently discussed separately. Understanding the relationship of these algorithms is valuable to the community, if correct.

**Weaknesses:**

Honestly, I couldn’t follow the analysis of the paper, which is probably on my side. Yet, I think several clarifications would be preferable to improve the paper.

- BoNBoN alignment is a combination of existing ideas, which is not a reason for rejection. The shortcoming of the paper is that it is unclear which part of the ideas is claimed to be their original as they do not cite the source of the ideas. 1. Using the output of the BoN as the alignment target is a common practice (Pace+ 2024; Liu+ 2023; Gulcehre+ 2023). 2. Mixing the SFT objective with the alignment objective is also proposed in the very first paper of RLHF (Ouyang+ 2022; PPO-ptx) 3. The disadvantage that DPO (or IPO) only controls the ratio of the chosen text/rejected text is commonly resolved by fine-tuning the model on the chosen response first, and then running the preference optimization using the pairs of the responses (Rafailov+2023). The novelty of the proposed algorithm would be clear if the origin of the ideas were clarified.
- I failed to follow the analysis of the paper. I would say that it needs some clarification to be understandable for a wide range of audiences.

Below is the comment to the paper assuming that I understand the argument of the paper correctly.
My concern with the paper is that it is making an unrealistic assumption and getting to the wrong conclusions. I suspect that assuming the functions are continuous does not simplify the argument. For example, if I understand it correctly, Lemma 5 is true only because we assume that y is completely ordered with r(x, y) and r is a one-to-one mapping. The assumption is not made for the sake of simplifying the argument. It is exploited to derive the results that are not valid without the assumption. If an assumption is required then it should be stated so. I would need a better explanation of why the assumption can be true.

**Questions:**

- Why do we assume pi_0 is continuous? It is discrete. There is an analysis of BoN that treats pi as a discrete function (Beirami et al. '24; Theoretical guarantees on the best-of-n alignment policy). What’s the advantage of assuming what is not true over the analysis which treats it as is?
- Eq. 3.1. I failed to understand this equation. My guess is that the right-hand side is ignoring the square of pi_0? If so, it should be explained. Is the := saying that we assume the situation where BoN policy can be represented so, or is it saying that the BoN policy defined in the preliminary can be described in this form?
- Eq. 3.5.: I believe this formula can be translated as Eq. (2) in (Liu et al. '24; Decoding-time Realignment of Language Models)". Is that correct?
- Line 143: “However, given the vast number of possible responses, the assumption is mild in practice.” Why is it true? Even if the domain is infinite, it does not mean that its support is large. One can think of a prompt from a closed QA that a model will most likely output only A or B.
- Theorem 2. and footnote 2: What does this Theorem claim? In the footnote, it says “the KL divergence is almost its upper bound log(n) - (n-1)/n”. But in Theorem 2 it says the KL divergence is *exactly* that value. Is it showing the upper bound or the exact value?
- Line 449: where the second to last equation is because… → I guess this refers to the fourth to the fifth equation? I guess this Z refers to Z^C_r(x)?
- Line 500: The approximation is due to the fact that p_i is small → Why is p_i small? We can think of closed QA tasks where the support of the language model is only A, B, C, and D. In this case square of p_i is not negligible.

**Limitations:**

I couldn't follow the discussion of the paper to the point where I could evaluate the limitations of the paper.

---

> ### Author Rebuttal · Authors · 2024-08-06
>
> Thank you for your review!
> - With respect to prior work: could you expand on the set of references you have in mind? E.g., with links or paper titles. It is not clear to us what papers you’re envisioning, or what connections you have in mind.
> - For RLHF (and DPO), it is standard procedure to first run SFT and then, on this SFT’d model, run the RLHF (DPO) procedure. Fundamentally, and most importantly, nothing in the standard RLHF (or DPO) pipeline targets the best-of-n distribution, which is the main point of the present paper. Additionally, the sense in which BonBon combines an SFT and contrastive objective is fundamentally different. The point in the present paper is that both of these objectives have the same analytic solution, so that the SFT time can be used *at contrastive alignment time* in the second step.
> - With respect to fine tuning on best-of-$n$ samples: a major contribution of the present paper is to show that the best-of-$n$ heuristic in fact has a remarkably crisp justification. As far as we know, this has not been previously clearly understood. We also note that we experiment with simply fine tuning on best-of-$n$ examples as a baseline and, as we explain at length, we find this works very poorly. The fact that it’s possible to design a contrastive procedure to explicitly target the best-of-$n$ distribution, and that this is much more effective than naive SFT, is an important contribution.
> - With respect to more minor comments:
>    - We adopt the continuous assumption for the following two reasons:
>      1. Although response $y$ exists in a high-dimensional space, making the assumption of continuity for $\pi_0$ appear unrealistic, the statistics we consider—KL divergence and win rate—are in a one-dimensional space. When the number of responses is large and each probability is low, the difference between the continuous and discrete cases becomes negligible. This point is well illustrated in Section B of the appendix.
>      2. We acknowledge that the continuous assumption may not be realistic for certain prompts, such as in Q&A scenarios. However, many tasks in LLM alignment aim to enhance abstract qualities like helpfulness or harmlessness of responses. In these settings, prompts often generate diverse responses without dominant answers, aligning with the assumptions mentioned in the previous point. Moreover, since the expectation is taken over a set $D$ of prompts $x$, we believe the theory remains relevant if most prompts in the set elicit diverse responses.
>
>      Note also that we find strong agreement with the best-of-$n$ theoretical predictions empirically, and that it is extremely rare in practice for two different responses to be assigned the same reward.
>    - Equation 3.1 is correct as written. The $:=$ denotes a definition, as is standard notation.
>    - It is not correct to say that equation 3.5 maps to Liu et al. These papers address different problems; they focus on maximizing the expectation of rewards, whereas our focus is on maximizing the win rate. Despite the similarity in the closed forms, they are not the same. More specifically, the difference is due to the fact that the expectation is also taken over $x$. Distributions of rewards $r(x,Y)$, $Y\sim\pi(y\mid x)$ can be different for different prompts.
>    - Re. Footnote 2. Assuming continuity makes the KL divergence larger than exploiting the discreteness in the actual responses. So, in this sense, the values we use here (derived under a continuity assumption) are upper bounds on the true discrete distribution. The point of this footnote is simply that in the case where the cardinality of response space is very large---as is typical---the gap between this “upper bound” and the analytic value is small. This is really just noting again that the continuity assumption is reasonable.
>    - Line 449: we believe this is correct as written.

---

> > ### Comment · Reviewer_PJN3 · 2024-08-08
> > **Thank you very much for the clarification**
> >
> > > With respect to prior work: could you expand on the set of references you have in mind? E.g., with links or paper titles. It is not clear to us what papers you’re envisioning, or what connections you have in mind.
> >
> > Sorry for the inconvenience. Here is the list of papers I mentioned.
> >
> > Pace+ 2024; West-of-N: Synthetic Preference Generation for Improved Reward Modeling https://arxiv.org/abs/2401.12086
> > Liu+ 2023; Statistical Rejection Sampling Improves Preference Optimization https://arxiv.org/abs/2309.06657
> > Gulcehre+ 2023; Reinforced Self-Training (ReST) for Language Modeling https://arxiv.org/abs/2308.08998
> > Ouyang+ 2022; Training language models to follow instructions with human feedback https://arxiv.org/abs/2203.02155
> > Beirami et al. '24; Theoretical guarantees on the best-of-n alignment policy https://arxiv.org/abs/2401.01879
> >
> > > For RLHF (and DPO), it is standard procedure to first run SFT and then, on this SFT’d model, run the RLHF (DPO) procedure. Fundamentally, and most importantly, nothing in the standard RLHF (or DPO) pipeline targets the best-of-n distribution, which is the main point of the present paper.
> > > With respect to fine tuning on best-of-n samples: a major contribution of the present paper is to show that the best-of- heuristic in fact has a remarkably crisp justification. As far as we know, this has not been previously clearly understood. We also note that we experiment with simply fine tuning on best-of-n examples as a baseline and, as we explain at length, we find this works very poorly. The fact that it’s possible to design a contrastive procedure to explicitly target the best-of-n distribution, and that this is much more effective than naive SFT, is an important contribution.
> >
> > I would say that Pace+ 2024; Liu+ 2023; Gulcehre+ 2023 can also be considered as targeting the best-of-n distribution. It would be helpful for the reader if the paper discussed the difference between the proposed method compared to the procedures in these papers as it is not immediate to me.
> >
> > > We adopt the continuous assumption for the following two reasons
> >
> > Thank you very much for the explanation. I think the assumption is valid. It would be helpful for the readers if it is clarified in the paper as the applications of LLMs are not constrained to open-ended text generation and they are also used in closed QA kinds of tasks where the possible output is limited (e.g., using it as a preference model to tell whether an answer A or B is preferred).
> >
> > > Equation 3.1 is correct as written
> >
> > I thought $\pi^{(n)}_{r}(y | x)$ will be $nQ_x(r(x, y))^{n-1} \pi_0(y | x) + \frac{n(n-1)}{2} Q_x(r(x, y))^{n-2} \pi_0(y | x)^2 + \frac{n(n-1)(n-2)}{6} Q_x(r(x, y))^{n-3} \pi_0(y | x)^3 + ... + \pi_0(y | x)^n$ as y can be sampled multiple times and we still get BoN policy to generate y. What am I missing?

---

> > > ### Author Response · Authors · 2024-08-09
> > >
> > > >I would say that Pace+ 2024; Liu+ 2023; Gulcehre+ 2023 can also be considered as targeting the best-of-n distribution. It would be helpful for the reader if the paper discussed the difference between the proposed method compared to the procedures in these papers as it is not immediate to me.
> > >
> > > The present paper makes two main contributions:
> > > (1) Theoretically, we established the connection between best-of-n sampling and other alignment methods, and proved that the best-of-n sampling distribution is essentially optimal with respect to win rate versus KL divergence.
> > > (2) Built upon the theoretical understanding, we propose BonBon as an efficient way to train a model to mimic its own best-of-n distribution.
> > >
> > > Compared to other works you mentioned:
> > > 1. Pace+ 2024 use the best-and-worst samples to further improve the reward model but not the language model. The theoretical results in this paper focus on the reward model as well. The papers are thus disjoint in their motivation and development.
> > > 2. Liu+ 2023 apply rejection sampling to get samples from $\pi_r(y\mid x)=\frac{1}{Z(x)}\pi_0(y\mid x)\exp\left(\frac{1}{\beta}r(x,y)\right)$ and then apply some contrastive method to to fine tuning. Their target policy is not the best-of-n policy.
> > > 3. Similarly, ReST essentially utilizes the samples from a reward-truncated reference model to do the fine-tuning. The target policy is also not the best-of-n distribution.
> > > 4. Compared to Ouyang+ 2022, they use reinforcement learning to do fine tuning and target at a different underlying policy.
> > > 5. The main point of Beirami et al. '24 is deriving KL divergence of best-of-n policy in the discrete case. We focus on understanding why best-of-n perform well and its connection to other alignment methods. We also discuss the KL divergence in the discrete case under a more general framework in addition to best-of-n policy.
> > >
> > > We emphasize that the referenced papers are wholly distinct from the present paper, both in motivation and results.
> > >  >Thank you very much for the explanation. I think the assumption is valid. It would be helpful for the readers if it is clarified in the paper as the applications of LLMs are not constrained to open-ended text generation and they are also used in closed QA kinds of tasks where the possible output is limited (e.g., using it as a preference model to tell whether an answer A or B is preferred).
> > >
> > > Thanks for your suggestions! We will expand the discussion where the assumption is introduced.
> > > >I thought $\pi_r^{(n)}(y|x)$ will be $nQ_{x}(r(x,y))^{n-1}\pi_{0}(y|x) + \frac{n(n-1)}{2}Q_{x}(r(x,y))^{n-2}\pi_{0}(y|x)^{2} + \frac{n(n-1)(n-2)}{6}Q_{x}(r(x,y))^{n-3}\pi_{0}(y|x)^{3} + \ldots + \pi_{0}(y|x)^{n}
> > > $ as $y$ can be sampled multiple times and we still get BoN policy to generate y. What am I missing?
> > >
> > > Since the reward model $r$ is a one-to-one mapping, you can get the best-of-n distribution by doing the integral:
> > > $$\pi_r^{(n)}(y\mid x) = \int n!\pi_0(y_1\mid x)\cdots\pi_0(y_{n-1}\mid x)\pi_0(y\mid x)1_{r(x,y_1)\le\cdots\le r(x,y_{n-1})\le r(x,y)}dy_1\cdots dy_{n-1}.$$
> > > This integral should be straightforward, noticing that $U_i=Q_x(r(x,Y_i))\sim U(0,1)$ with $Y_i\sim\pi_0(y\mid x)$.

---

> > > > ### Comment · Reviewer_PJN3 · 2024-08-11
> > > >
> > > > > We emphasize that the referenced papers are wholly distinct from the present paper, both in motivation and results.
> > > >
> > > > Thank you very much for the explanation. I am not concerned with the "novelty" of the paper. I am concerned that the distinction from the existing work is not explicitly stated in the paper. It is now clear what the point of the paper is.
> > > >
> > > > > Since the reward model $r$ is a one-to-one mapping, you can get the best-of-n distribution by doing the integral: $$\pi_r^{(n)}(y\mid x) = \int n!\pi_0(y_1\mid x)\cdots\pi_0(y_{n-1}\mid x)\pi_0(y\mid x)1_{r(x,y_1)\le\cdots\le r(x,y_{n-1})\le r(x,y)}dy_1\cdots dy_{n-1}.$$ This integral should be straightforward, noticing that $U_i=Q_x(r(x,Y_i))\sim U(0,1)$ with $Y_i\sim\pi_0(y\mid x)$.
> > > >
> > > > I see. Because we let the measure of y to be a continuous density function we can basically ignore the probability of sampling exactly the same value twice.
> > > > I would say that this is also not that straightforward to understand, at least for some of the readers including me especially because we have the formulation of Beirami et al. '24 in mind, so it would be helpful to have an explanation for it.

---

> > > > > ### Author Response · Authors · 2024-08-12
> > > > >
> > > > > >Thank you very much for the explanation. I am not concerned with the "novelty" of the paper. I am concerned that the distinction from the existing work is not explicitly stated in the paper. It is now clear what the point of the paper is.
> > > > >
> > > > > Thank you for the suggestions. We will expand the related works section to note the references you have suggested.
> > > > >
> > > > > >I see. Because we let the measure of $y$ to be a continuous density function we can basically ignore the probability of sampling exactly the same value twice. I would say that this is also not that straightforward to understand, at least for some of the readers including me especially because we have the formulation of Beirami et al. '24 in mind, so it would be helpful to have an explanation for it.
> > > > >
> > > > > We will add a note that this is what the continuity assumption does.

---

> > > > ### Public Comment · ~Yunyi_Shen1 · 2025-01-05
> > > > **Do you need r to be one-to-one?**
> > > >
> > > > Thanks a lot for the great paper and discussion. I came to the review page to see if there is a clarification on this one-to-one assumption that would imply natural languages are essentially one-dimensional objects. I wonder do you really need this assumption for 3.1 to be true? Let's pretend the reward $r$ is also random for now, but with a distribution condition on $x,y$ being $\delta$ measure. The joint distribution of $r,y$ condition on $x$ for the base model can be written as $\pi_0(y, r|x)=\pi_0(y|x)\pi_0(r|x,y)$ and the second factor is just $\delta$ measure since $r$ is fully determined by $x,y$. Another view is $\pi_0(y, r|x)=\pi_0(r|x)\pi_0(y|r,x)$. This will be useful for the order statistics result. A modified version of eq. 3.1 with $r,y$ can be written I think as $\pi^{(n)}(r,y)=nQ(r)^{n-1}\pi_0(r|x)\pi_0(y|r,x)$, the first factor $nQ(r)^{n-1}\pi_0(r|x)$ is just order statistics result on marginals of $r|x$ and the second factor of $y|r,x$ kept to be $\pi_0(y|r,x)$ because our best of n is on $r$ not $y$ directly. Now rewrite this $\pi^{(n)}(r,y)=nQ(r)^{n-1}\pi_0(y|x)\pi_0(r|y,x)$, and $\pi_0(r|y,x)$ is a $\delta$ measure, integrate $r$ out would get eq 3.1 back without the need of $r$ being one-to-one with $y$. Am I missing anything?

---

> > > > > ### Public Comment · ~Lin_Gui5 · 2025-01-05
> > > > >
> > > > > Thanks for your comment. Yes, the one-to-one mapping is not necessary and what we essentially need for eq 3.1 is that $r(x,Y_0)$, $Y_0\sim\pi_0(y\mid x)$ is a continuous random variable.

---

### Official Review · Reviewer_Jgyb · 2024-07-15

**Soundness:** 3
**Presentation:** 4
**Contribution:** 3
**Rating:** 7
**Confidence:** 4

**Summary:**

This paper addresses aligning samples from large language models (LLMs) with human preferences using best-of-$n$ sampling, which involves drawing $n$ samples, ranking them, and selecting the best one. It tackles two main problems. First, it explores the relationship between best-of-$n$ sampling and Reinforcement Learning from Human Feedback (RLHF) approaches. The authors demonstrate that the best-of-$n$ sampling distribution is essentially equivalent to the RLHF policy when a specific monotone transformation is applied to the reward function. This transformation optimizes the trade-off between win-rate against the base model and KL distance from the base model, making best-of-$n$ a Pareto-optimal solution for win-rate vs. KL distance. Second, the paper introduces BonBon Alignment, a method to fine-tune models to mimic the best-of-$n$ sampling distribution, thus avoiding the need to draw $n$ samples for each inference. Experiments indicate that BonBon Alignment yields models with high win rates while minimally impacting off-target aspects of the generations.

**Strengths:**

1. This paper is well written. The notations are clear and the literature review is sufficient.
2. By justifying that the best-of-n policy is essentially optimal in terms of win rate versus KL-divergence, efficient training for language models based on best-of-n fine tuning is achieved by mimicking the best-of-n sampling distribution.
3. The control of hyperparameter is made simpler with $\alpha$, that balances the loss ingredients.

**Weaknesses:**

When multiple aspects of human preference exist, the proposed method seems to have limited capacity to handle contradictory preference ratings, e.g., helpfulness and harmfulness. Therefore, because the trade-off between diverse aspects of preferences is not explicitly addressed.

**Questions:**

With an only $\alpha$ controling the divergence, it would be interesting to understand how could the proposed Bonbon alignment reflect multiple aspects of real world preferences, e.g., the balance between helpfulness and harmlessness in Anthropic dataset.

---

> ### Author Rebuttal · Authors · 2024-08-06
>
> Thank you for your review!
>
> With respect to multiple aspects: we agree this is a fundamental challenge with preference modeling, and a very interesting subject for future research. We note, however, that this problem is essentially fundamental to all post-training procedures. E.g., even explicit reward modeling has to define a way of aggregating multiple distinct kinds of reward. Any aggregation scheme would then induce a preference labeling. And our results would apply to this preference labeling.
>
> That is: the question of _how to rank samples_ and _what to do with the rankings_ are separable questions. The present paper addresses the second problem. Progress on the first is also very interesting, but is out of scope for this paper.

---

> > ### Comment · Reviewer_Jgyb · 2024-08-12
> >
> > I have read the rebuttal and thank the authors for their candid responses.
> >
> > I maintain my positive opinion on this paper.

---

### Author Rebuttal · Authors · 2024-08-06

We thank the reviewers for their insightful comments and constructive submissions. Where appropriate, we have incorporated these into the main text (details in reviewer-specific replies), and we believe this has strengthened the paper.

The reviewers agree that the paper addresses an important and interesting problem (PJN3, mr5a), is clearly-written (Jgyb, ySG3), theoretically sound (Reviewer mr5a, ySG3), and with solid experimental results (Reviewer Jgyb, mr5a).

---

### Decision · Program_Chairs · 2024-09-25

**Decision:**

Accept (poster)

**Comment:**

The paper studies the connection between best-of-n-based fine-tuning and traditional RLHF methods. Inspired by the effectiveness of best-of-n, the authors also propose a new algorithm for better performance. The analysis and experimental results are interesting. I recommend acceptance.